# Psychometric Properties of the Emotional Eater Questionnaire in University Students

**DOI:** 10.3390/ijerph191710965

**Published:** 2022-09-02

**Authors:** Elena Sosa-Cordobés, Francisca María García-Padilla, Ángela María Ortega-Galán, Miriam Sánchez-Alcón, Almudena Garrido-Fernández, Juan Diego Ramos-Pichardo

**Affiliations:** Department of Nursing, University of Huelva, 21071 Huelva, Spain

**Keywords:** Emotional Eater Questionnaire, exploratory factor analysis, confirmatory factor analysis, college students, emotional eating, obesity

## Abstract

Emotional Eating (EE) patterns have been shown to play a relevant role in the development of overweight and obesity. The aim of this study was to analyze the factor structure and psychometric properties of the Emotional Eater Questionnaire (EEQ) in university students from Huelva. The EEQ was administered to 1282 students (age 22.00 (±5.10), BMI 23.59 (±6.74)), belonging to the University of Huelva. An exploratory factor analysis (EFA) and confirmatory factor analysis (CFA) were carried out. The internal structure of the questionnaire, internal consistency, test-retest reliability, and convergent validity were analyzed. Principal component analysis of the questionnaire showed two dimensions, explaining 56% of the variance. Internal consistency showed a Cronbach’s alpha of 0.859 globally, and of 0.841–0.855 if the items were removed. The corrected item-total correlation yielded values of 0.444–0.687. The test-retest stability was ICC = 0.924 (*p* < 0.001). The data showed significant correlations between EEQ and the rest of the variables, and a Spearman’s Coefficient ranging from −0.367 to 0.400. The fit indexes were good for the confirmatory factor analysis. The results obtained with this structure found an adequate reliability and validity of the questionnaire in comparison with previous studies.

## 1. Introduction

The university population is a large group, generally between 18 and 24 years of age, in a state of rapid physical growth and mental development. University life favours the development of unhealthy habits that, if not modified, may lead to an increase in mortality in the future [1].

The transition period from adolescence to adulthood is characterized by an increased risk of acquiring habits that are harmful to one’s health: inadequate nutrition, sedentary lifestyles, consumption of toxic substances, risky sexual behaviour, etc. Among the factors responsible are an increase in independence and responsibility for self-care, managing increasingly frequent stressful situations ineffectively, and greater vulnerability to peer influence [2,3].

In fact, there is a widely known term, the ‘Freshman 15’, used in the United States to refer to an arbitrarily set amount of 15 pounds (7 kg) (originally only 10 pounds (5 kg)) of weight gained during the first year of university, largely due to these changes in habits. In Australia and New Zealand, they are sometimes referred to as ‘First Year Fatties’, ‘Fresher Spread’, or ‘Fresher Five’, referring to the 5 kg gain [4].

Some studies have found that stress can trigger emotional eating behaviours, meaning that people living in more stressed societies are more likely to be obese [5]. Furthermore, the 2019 coronavirus pandemic (COVID-19) and the consequent mandatory quarantine increased symptoms of mental disorders, and inferentially, emotional eating (EE) [6,7,8]. Emotional eating refers to the action of eating food because of emotions. Although EE was originally defined as eating in response to negative emotions, there are now a number of studies showing that a positive mood can also lead to increased food intake [9,10,11]. Eating in response to emotions can lead to problems such as an increased body mass index (BMI), interference with weight loss and weight maintenance, stress, anxiety, binge eating, and depression, among other consequences [10]. Evidence suggests that the problem is not necessarily associated with the experience of emotions per se, but rather with a lack of adaptive emotion regulation strategies [12].

Estimates of the prevalence of EE vary across different studies. Approximately 20–45% of non-clinical adult samples are identified as emotional eaters [13]. Among overweight adults, rates approach 60%. Although obesity is considered to be the result of a variety of interactions between several factors (genetic, socioeconomic, endocrine, metabolic, and psychological), research suggests that EE is quite common among overweight individuals, who score higher on EE measures than individuals within the normoweight category [11,13,14,15].

Youth obesity is associated with an increased likelihood of obesity, premature death, and disability in adulthood. High body mass index is an important risk factor for non-communicable diseases, such as cardiovascular disease, diabetes, osteoarthritis, and some cancers [16].

EE, although paradoxical, is common among many people; those with emotional eating problems do not always recognize it as such and, thus, they do not seek help [5]. Research on EE, including examining the mechanisms involved, and determining its clinical consequences requires the development and validation of psychometric scales to identify and quantify the construct [13].

In eating research, it is common practice to group people into different types of eaters, such as emotional, external, and restrained eaters. This categorization is usually based on scores from self-report questionnaires [17]. There are different scales to assess emotions and other non-traditional factors contributing to overweight and obesity such as the Emotional Eating Scale (EES) [18], the Dutch Eating Behaviour Questionnaire (DEBQ) [19], the Mindful Eating Questionnaire (MEQ) [20], the Emotional Overeating Questionnaire (EOQ) [21], the Three Factor Eating Questionnaire (TFEQ) [22], and the Salzburg Stress Eating Scale (SSES) [23].

Unfortunately, some of these questionnaires are designed to assess other eating disorders, are specific to obesity, are too long, or are too complicated to be applied in everyday practice. Some of these questionnaires have been translated, adapted, and validated for the Spanish population. However, very few validation studies have been conducted to determine the reliability and validity of each instrument in specific population groups [13].

The Emotional Eating Questionnaire (EEQ) was developed and validated less than a decade ago, being the first psychometric measure of EE developed in Spanish. Due to its relevance and brevity, the EEQ could facilitate the early identification of the level of EE, encourage early and appropriate intervention and, consequently, reduce the future prevalence of overweight and obesity in the university population [24].

It has been used to assess EE in a Spanish population with binge eating disorder. The validation of this questionnaire was carried out in 354 participants with obesity, all of them undergoing dietary and behavioural treatment for weight loss by Garaulet et al. After principal components analysis, the questionnaire was classified in three different dimensions that explained 60% of the total variance: disinhibition, type-of-food and guilt. Internal consistency tests found that Cronbach’s alpha was 0.773 for the disinhibition subscale, 0.656 for the type-of-food subscale and 0.612 for the guilt subscale. The test-retest stability was r = 0.70. The data showed that the percentage of agreement between the EEQ and the MEQ was around 70%, with a Kappa index of 0.40; *p* < 0.001 [25].

It has been adapted for Chilean university students by Mariela et al. A good item-test correlation was found, the factorial structure was similar to the original questionnaire, and it had good internal consistency [26]. In Spain, the metric properties of the EEQ for university students have been evaluated in a study with a sample of 295 participants, in which neither test-retest reliability nor confirmatory factor analysis (CFA) data are provided; only exploratory factor analysis has been done (Spain) by Bernabeu et al. [13]. Therefore, it is extremely necessary to carry out a CFA in Spanish university students in order to be able to use it as evidence for this population.

The aim of this study was to evaluate the metric properties in a sample of university students, including test-retest reliability and CFA.

Its implementation in a university population will deepen the understanding of the mechanisms involved in EE, as well as allow researchers to design and develop strategies for the prevention and treatment of overweight and obesity.

## 2. Materials & Methods

### 2.1. Participants

Participants were 1282 students from the University of Huelva, a small town on the west coast of Spain. The inclusion criteria were that the participants had to be enrolled in the 2019/2020 academic year and degree, be present in the classroom on the day of recruitment, and complete the entire questionnaire, or at least all the mandatory fields. In addition, they had to voluntarily agree to participate in this study. Students who were abroad on an academic exchange at the time of data collection were excluded. The sample was recruited by inviting all students to participate in the study. Permission was obtained from the university to contact faculty directors and degree coordinators regarding student access. Teachers were then contacted for permission for researchers to visit classrooms or join Zoom sessions to recruit students for the study.

The students participated voluntarily and anonymously without receiving any compensation. Furthermore, students were informed that their participation was not linked to the assessment of any subject. Data were always treated anonymously and according to the principles of the Declaration of Helsinki. The study was approved by the Research Ethics Committee of the Andalusian Public Health System in Huelva (MINPE2020).

### 2.2. Instrument

The multidimensional nature of eating behaviour means that its assessment is not easy. The EEQ is a scale on EE that is easy to apply in healthcare practice. The EEQ classifies obese individuals according to the relationship between food intake and emotions. It consists of ten items that assess the extent to which they affect eating behaviour. All questions have four possible answers: (1) Never; (2) Sometimes; (3) Usually; and (4) Always. Each answer is given a score from 0 to 3, where the lower the score, the healthier the behaviour. For clinical practice, subjects are classified into four groups according to the score obtained. For a score between 0–5: non-emotional eater; for a score between 6–10: somewhat emotional eater; for a score between 11–20: emotional eater; and for a score between 21–30: highly emotional eater [24].

The original principal components analysis of the questionnaire found three different dimensions explaining 60% of the variance: disinhibition, type-of-food, and guilt. Internal consistency tests found that the Cronbach’s alpha was 0.773 for the disinhibition subscale, 0.656 for the food type subscale, and 0.612 for the guilt subscale. The test-retest stability was r = 0.70. The data showed that the percentage of agreement between the EEQ and MEQ was 70%, with a Kappa index of 0.40, *p* < 0.001, indicating adequate reliability of the instrument [24].

The profile of the student’s health status was investigated using the SF-36 health questionnaire, and the EQ-5D. The SF-36 is made up of 36 questions (items) that assess both positive and negative states of health. The 36 items of the instrument cover the following scales: physical function, physical role, body pain, general health, vitality, social function, emotional role, and mental health. All items are scored so that a high score represents a more favourable health status. As well, each item is scored from 0 to 100, so the lowest and highest possible scores are 0 and 100, respectively. The ratios represent the percentage of the total possible score achieved. Internal consistency tests found alphas between 0.83 and 0.93 for the eight scales, and 0.94 and 0.89 for the physical (PCS) and mental (MCS) component summary measures [27], respectively. The Spanish version of the Short Form 36 Health Survey had a reliability higher than the suggested standard (Cronbach’s alpha) of 0.7 in 96% of the evaluations. Grouped evaluations obtained by meta-analysis were higher than 0.7 in all cases [28].

The EQ-5D descriptive system comprises the following 5 dimensions: mobility, self-care, usual activities, pain/discomfort, and anxiety/depression. Each dimension has 3 levels: no problems, some problems, and extreme problems. This is transformed into a 1-digit score that expresses the level that corresponds with a particular dimension. The score for the five dimensions can then be combined into a 5-digit score that represents the patient’s health state. The EQ-5D had good discrimination for diagnosed diseases (ranging from 64.3% to 86.3%). Floor/ceiling effects were observed across all items. The EQ-5D-3L’s total score discriminated between respondents [29]. The Spanish version of the Short Form 36 Health Survey had a good reliability, higher than 0.7 in 95% of the evaluations. Grouped evaluations obtained by meta-analysis were higher than 0.7 in all cases [30].

Regarding mental health, the Hospital Anxiety and Depression Questionnaire (HADS) was included. This scale is an instrument that is made up of two subscales (HADA: anxiety, and HADD: depression) made up of seven items each, with scores that range from 0 to 3. The authors recommend the original cut-off points: eight for possible cases, and >10 for probable cases on both subscales. The two HADS subscales (anxiety and depression) had excellent internal consistencies (Cronbach’s α value of 0.82–0.83), and a factor analysis confirmed a two-factor structure. The convergent validity of the HADS subscales appeared to be good due to the significant correlations between HADS and MSIS-29 [31]. Results showed an optimal internal consistency and test-retest reliability (>0.70). The validity with other anxiety and depression subscales showed a good correlation. Sensibility and specificity were satisfactory when using the HAD total scale for adults (>0.80) More research was needed with respect to the HAD’S global psychological distress scale in different Spanish samples [32]. Data were entered into a database for analysis using SPSS^®^ version 25 software [33].

### 2.3. Data Collection

Students completed a set of self-administered questionnaires at the beginning or at the end of the class sessions, with the teacher’s permission. The questionnaires were provided online by a researcher after receiving the participant’s informed consent. Completion of the questionnaires took about 20 min. In addition, to avoid falsification of data we excluded participants whose ages did not match their birth dates, and checked that participants had not answered in an “All or Nothing” manner of responding or skipped large portions of the survey. In addition, to manage the data and avoid data falsification or duplicate responses, they were asked at the beginning of the questionnaire for a code made up of the last four digits and the letter of their ID. Finally, the link to the survey was sent directly to the students at the time the researcher was in the corresponding class, and the survey was programmed so that it could only be answered once; the submitted answers were not allowed to be modified. Participants completed self-reported information about their age, weight, and height. Subsequently, participants filled in the EEQ, the Short-Form 36 (SF-36), the European Quality of Life Five Dimension Scale (EQ-5D), and the Hospital Anxiety and Depression Scale (HADS).

### 2.4. Data Analysis

Means and standard deviations of item scores were calculated. Ceiling and floor effects were determined by considering that either of these existed when the percentage of clustered responses at the highest or lowest item value was greater than or equal to 15% [27]. For this purpose, the proportion of respondents with the lowest or highest possible score was described.

For the determination of internal consistency, Cronbach’s alpha was calculated for the overall scale and for each dimension, as well as the corrected item-test correlations. Test-retest reliability was assessed with the intraclass correlation coefficient (ICC) for a subsample of 152 students at an interval of 2.3 weeks. Many students were asked to complete the questionnaire again, and those 152 participants who were anonymously identified by code as questionnaire responders in both periods were included in the test-retest sample.

To study the factor structure of the questionnaire, an exploratory factor analysis was initially carried out using the maximum likelihood (ML) method with oblimin rotation, considering factors with eigenvalues greater than 1. Subsequently, a confirmatory factor analysis was performed using a robust estimation method and three factor-structure models were compared: the three-factor model proposed by the original authors of the scale (model 2), the two-factor model proposed by Bernabéu et al. [13] (model 3), and the model identified for this study in the previous EFA, also with two factors, but with item 9 included in another factor.

For each of the three models, the comparative fit index (CFI) and the incremental fit index (IFI) were calculated, with values >0.9 indicating an acceptable fit and >0.95 indicating a good fit [34]. The root mean square error of approximation (RMSEA), which favours more parsimonious models, with a value ≤0.06 indicating a good fit, and ≤0.10 representing an acceptable fit. Finally, the χ^2^/DF was also calculated and considered acceptable if <4 [35,36]. EQS 6.1 software [37] was used for confirmatory factor analysis. Other analyses were performed with SPSS 25.0 [33].

## 3. Results

### 3.1. Description of the Sample

The sample consisted of 1282 students from the University of Huelva (Spain). Students from all years, from 25 different university degrees were included. Table 1 shows descriptive information about the sample studied. 25.8% were overweight or obese, 66.6% were emotional eaters, 46.3% had symptoms of anxiety, and 18% had symptoms of depression. The lowest scoring subcategories of the SF36 were SF36VT (vitality), SF36MH (mental health), SF36GH (general health), and SF36RE (role emotional). The SF36VT is a general measure of energy/fatigue; the SF36MH, which was designed to measure general mental health status, includes four major mental health dimensions: anxiety, depression, loss of behavioural/emotional control, and psychological well-being; the SF36GH measures the general and subjective state of the patient with respect to others and over time; and the SF36RE measures role limitations due to mental health difficulties with three items, including the amount of time spent on work or other activities, the amount of work accomplished, and the care with which work is performed [38].

There were significant differences between the sexes for all variables except for the HADS-D (Hospital Anxiety and Depression Scale Depression subscale), a subscale which assesses depression [31] (Table 1).

### 3.2. Descriptive Analysis of Each Item

Table 2 shows descriptive data at the item level. The observed range of means was from 0.35 to 1.41 (with scores ranging from 0 to 3). The students answered 100% of each item. Participants had the lowest score on item 9 and the highest score on item 2. All items showed a positive skew. Two items had a leptokurtic distribution (items 9 and 10), while eight items had a platykurtic distribution (items 1, 2, 3, 4, 5, 6, 7, and 8). There was a floor effect for all items except for item 2 (*p* < 0.001). The floor effect is what happens when there is an artificial lower limit, below which data levels cannot be measured. This can be caused when there is a lower limit on a survey or questionnaire, and a large percentage of respondents score near this lower limit.

### 3.3. Validity

#### 3.3.1. Convergent Validity

Table 3 shows the Spearman correlation between the EEQ (and F1–F2) and the rest of the variables measured. These correlations were statistically significant. The level of emotional eating was correlated with anxiety status, depression, and BMI. The correlations were significant with all variables and ranged from −0.367 to 0.400. The highest correlations were related to emotional state: −0.367 (SF36MH), −0.306 (SF36RE), 0.400 (HADS-anxiety), and 0.360 (HDAS-depression). Both F1 and F2 had the highest correlations with SF36 total, mental health, and anxiety.

#### 3.3.2. Exploratory Factor Analysis

Bartlett’s statistics and KMO tests showed the adequacy of the polychoric correlation matrix to the factor model (χ^2^ (45) = 4316.769; *p* < 0.001; KMO = 0.901). Table 4 shows the factor extraction by the eigenvalue-based ML method. Two factors with eigenvalues greater than 1 were identified, which together explained 56% of the variance (45% for the first factor and 11% for the second). The factorial weight suggested a first factor composed of 6 items (2, 3, 4, 5, 6, and 8) and a second factor composed of 4 items (1, 7, 9, and 10). The variance of the 2 factors with eigenvalues >1 was 56% (44.7% for the first factor and 11.3% for the second).

#### 3.3.3. Confirmatory Factor Analysis

As shown in Table 5, this structure met all the values recommended by H. & Bentler [36]. The CFI and IFI indexes were higher, and the RMSEA lower for our model than for the other two. Figure 1 describes the model proposed. This was made up of two factors: impulsive eating and mood/affect. Factor 1 was made up of the items: 2, 3, 4, 5, 6, and 8. Factor 2 was made up of the items: 1, 7, 9, and 10. In addition, the figure also reflects residual errors (E1–E10).

Model 3: Standardized factor weights and residuals. Figure 1 legend: F1: Impulsive Eating; Factor 2: Mood/Affect. E1–E10 refer to the residual errors of the observed scores of each item; this is an expression of the variance not explained by the latent variables.

### 3.4. Reliability

#### 3.4.1. Internal Consistency

The overall Cronbach’s alpha was 0.859. The alpha for our proposed model was 0.805 for dimension 1, and 0.787 for dimension 2. The corrected item-test correlations were between 0.444 for item 8, and 0.687 for item 10. The item contribution to internal consistency ranged from 0.835 to 0.855 if each item was removed (Table 6).

#### 3.4.2. Test-Retest

The ICC value showed excellent reliability between the first and second administration of the EEQ (ICC = 0.924; *p* < 0.001).

## 4. Discussion

The Emotional Eating Questionnaire is the first psychometric measure of EE developed in Spanish. Emotions have a powerful effect on our choice of food and eating habits. It has been found that in some people there is relationship between eating, emotions, and an increased energy intake. This relationship should be measured to better understand how food is used to deal with certain mood states, and how these emotions affect the effectiveness of weight loss programs [39].

The aim of the study was to analyze the factor structure and psychometric properties of the EEQ in a sample of Spanish university students. The EEQ is a short, easy-to-administer questionnaire.

The EEQ had good psychometric properties in the university population of Huelva (Spain). This indicated that the application of the present questionnaire seemed to help in identifying university students with emotional eating.

The results suggested some differences in the factor structure of the EEQ in the university population of Huelva in comparison with previous studies conducted on a clinical population [24], university students in Chile [26], and university students in Tarragona (Spain) [13] (see Table 1). These dissimilarities could be explained by the differences in the sample between the original one, which only included overweight or obese people [24], and the rest, which included university students [13,26]. On the other hand, the differences between the Tarragona study [13] and the present one may be due to the differences between samples in terms of size, sex distribution, degree and year, % of people with overweight or obesity, etc.

The main difference between [24] study and [13] and our study is the number of factors: 3 versus 2. Garaulet describes 3 subscales: disinhibition, type-of-food, and guilt. Bernabeu describes the first factor one as those behaviours most directly related to emotional eating, and the second factor as the emotional consequences of the lack of control over food intake. The present study labeled the first factor as impulsive eating, and the second factor as mood/affect.

In addition, differences in design may also have led to different results. Bernabéu et al. [13] used a sample of 295 students from three degrees, while for the present study, 1282 students from 25 degrees were included. In addition, the test-retest reliability test and the CFA were added. Finally, Bernabéu et al. [13] performed the convergent validity analysis with the STAI, BMI, Healthy Lifestyle report, Healthy Food Consumption report, Awareness About Food Intake report, and Perception About Weight, and the present study used The Short-Form 36 (SF-36), the European Quality of Life Five Dimension Scale (EQ-5D), and the Hospital Anxiety and Depression Scale (HADS).

The results of this study largely confirm previous validations, but they need to be interpreted with caution, as the original study was conducted with clinical people of all ages, while this one also included individuals from a non-clinical population and was limited to the university students (of a younger age). In addition, the present study did not differentiate by sex, a variable that could affect the interpretation of the results.

### Limitations

Even though these limitations exist, the present study provides additional validation to the recent Emotional Eater Questionnaire. However, weight and height were self-reported information, and this could influence the results; some studies have examined the accuracy of self-reported versus directly measured height and weight, but findings varied, and many studies were small or otherwise limited. Current weight has been shown to influence weight reporting accuracy. The overweight and obese tend to under-report their weight and the underweight tend to over-report [40].

Given the significant differences between almost all variables and sex, the Item Response Theory (IRT) model should be applied in the future to the development, evaluation, and refinement of the questionnaire in order to construct or adapt the instrument with properties that do not vary between populations, so that both women and men are equally likely to give the same response [41]. Further exploration among non-clinical Spanish samples would also be beneficial, as EE models may differ between student samples and the general population. Finally, although there is currently a large body of evidence on EE, this construct is not as simple as it appears to be. There is a lack of solid theoretical frameworks on the behaviour of EE and on the mechanisms involved, which, in turn, hinders the interpretation of the results [10].

## 5. Conclusions

In conclusion, a new validation of the Emotional Eater Questionnaire has been presented for a Spanish university population, resulting in a valid and reliable questionnaire to assess the degree of emotional eating. The early identification of this eating pattern in young people will allow for the design of personalized interventions and strategies from the beginning of the university stage.

## Figures and Tables

**Figure 1 ijerph-19-10965-f001:**
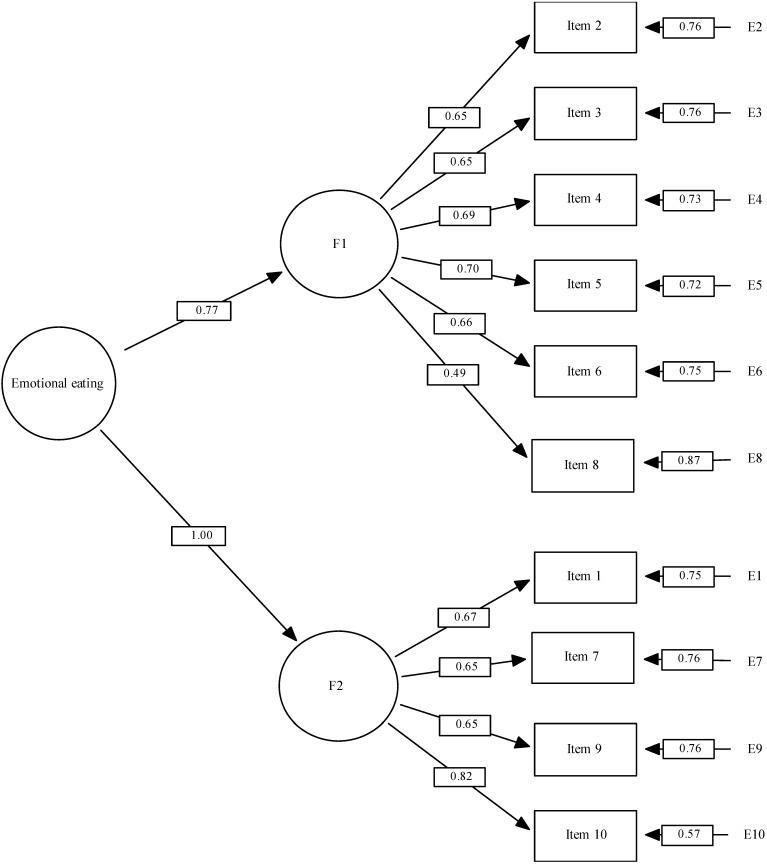
Factor structure of the proposed model for the Emotional Eater Questionnaire.

**Table 1 ijerph-19-10965-t001:** Description of the studied sample.

Variables	Males	Females	Chi-Square or *t*-Test *p*-Values	Total Sample
Age	22.44 (sd = 5.40)	21.76 (sd = 4.91)	2.27 (0.024)	22.00 (sd = 5.10)
BMI	24.06 (sd = 5.75)	23.16 (sd = 5.35)	2.79 (0.005)	23.59 (sd = 6.74)
Low weight	4.46%	6.83%		6% (*n =* 77)
Normoweight	63.62%	70.50 %		68.2% (*n =* 874)
Overweight	26.56%	14.51%		18.7% (*n =* 240)
Obesity I	2.23%	2.28%		4.8% (*n =* 62)
Obesity II	4.24%	5.76%		2.3% (*n =* 29)
EEQ	6.96 (sd = 4.81)	9.75 (sd = 5.85)	−8.64 (<0.001)	8.77 (sd = 5.66)
Not emotional	44.87%	27.22%		33.4% (*n =* 428)
Somewhat emotional	36.16%	33.81%		34.6% (*n =* 444)
Emotional	17.63%	33.57%		27.8% (*n =* 356)
Highly emotional	2.01%	5.04%		4.2% (*n =* 54)
HADS-A	6.75 (sd = 3.92)	8.39 (sd = 4.42)	−6.60 (<0.001)	7.81 (sd = 4.32)
No case	64.29%	48.08%		53.7% (*n =* 689)
Doubtful case	19.20%	19.30%		19.3% (*n =* 247)
Case	16.52%	32.61%		27% (*n =* 346)
HADS-D	4.23 (sd = 3.33)	4.49 (sd = 3.43)	−1.28 (0.202)	4.40 (sd = 3.40)
No case	83.48%	88.01%		82% (*n =* 1051)
Doubtful case	12.28%	6.12%		12.5% (*n =* 161)
Case	4.24%	5.88%		5.5% (*n =* 70)
EQ-5D	0.89 (sd = 0.15)	0.85 (sd = 0.16)	4.09 (<0.001)	0.86 (sd = 0.16)
SF36 Total	76.77 (sd = 12.20)	71.05 (sd = 13.16)	7.61 (<0.001)	73.05 (sd = 13.11)
SF36PF	96.15 (sd = 6.64)	93.23(sd = 10.32)	5.42 (<0.001)	94.25 (sd = 9.31)
SF36RP	85.46 (sd = 17.90)	82.01 (sd = 18.61)	3.21 (0.001)	83.22 (sd = 18.43)
SF36BP	80.15 (sd = 20.13)	74.50 (sd = 21.53)	4.58 (<0.001)	76.47 (sd = 21.21)
SF36GH	68.36 (sd = 16.39)	62.34 (sd = 16.42)	6.26 (<0.001)	64.45 (sd = 16.65)
SF36VT	61.13 (sd = 18.41)	54.10 (sd = 19.03)	6.47 (<0.001)	56.50 (sd = 19.11)
SF36SF	80.66 (sd = 21.62)	73.88 (sd = 23.75)	5.03 (<0.001)	76.25 (sd = 23.24)
SF36RE	73.87 (sd = 23.31)	66.54 (sd = 22.32)	5.52 (<0.001)	69.10 (sd = 22.92)
SF36MH	68.39 (sd = 18.79)	61.89 (sd = 18.71)	5.93 (<0.001)	64.16 (sd = 18.98)

Note: SF36 Physical functioning; SF36 Role physical; SF36 Bodily pain; SF36 General health; SF36 Vitality; SF36 Social Functioning; SF36 Role emotional; SF36 Mental Health; European Quality of Life-5 Dimensions’ Coefficient; Anxiety subscale of the HADS questionnaire; Depression subscale of the HADS questionnaire; Body mass index. (*p* < 0.001).

**Table 2 ijerph-19-10965-t002:** Descriptive analysis of each item, as well as floor and ceiling effects.

Item	$\bar{X}$	SD	Asymmetry	Kurtosis	Floor	Ceiling
1	0.88	0.939	0.909	−0.052	41.5%	9.2%
2	1.41	0.768	0.640	−0.090	6.2%	11.2%
3	0.79	0.901	1.004	−0.187	46.6%	6.9%
4	0.80	0.811	0.883	−0.374	40.1%	4.5%
5	1.34	0.932	0.390	−0.686	17.4%	14.8%
6	1.01	0.940	0.609	−0.561	35.3%	8.6%
7	0.95	0.907	0.479	−0.846	38.8%	4.8%
8	0.63	0.804	1.159	−0.685	54.1%	3.4%
9	0.35	0.690	2.160	4.472	74.1%	2.7%
10	0.61	0.810	1.341	1.331	54.8%	4.7%

**Table 3 ijerph-19-10965-t003:** Spearman correlation coefficients between EEQ and measures used for convergent validity.

Measures	EEQ	F1 (Items: 2, 3, 4, 5, 6, 8)	F2 (Items: 1, 7, 9, 10)
SF36 Total	−0.398 (*p* < 0.001)	−0.368 (*p* < 0.001)	−0.332 (*p* < 0.001)
SF36PF ^a^	−0.204 (*p* < 0.001)	−0.179 (*p* < 0.001)	−0.165 (*p* < 0.001)
SF36RP ^b^	−0.207 (*p* < 0.001)	−0.165 (*p* < 0.001)	−0.174 (*p* < 0.001)
SF36BP ^c^	−0.196 (*p* < 0.001)	−0.202 (*p* < 0.001)	−0.140 (*p* < 0.001)
SF36GH ^d^	−0.271 (*p* < 0.001)	−0.252 (*p* < 0.001)	−0.204 (*p* < 0.001)
SF36VT ^e^	−0.326 (*p* < 0.001)	−0.310 (*p* < 0.001)	−0.291 (*p* < 0.001)
SF36SF ^f^	−0.284 (*p* < 0.001)	−0.270 (*p* < 0.001)	−0.261 (*p* < 0.001)
SF36RE ^g^	−0.306 (*p* < 0.001)	−0.305 (*p* < 0.001)	−0.247 (*p* < 0.001)
SF36MH ^h^	−0.367 (*p* < 0.001)	−0.331 (*p* < 0.001)	−0.341 (*p* < 0.001)
EQ5D Coeff. ^i^	−0.256 (*p* < 0.001)	−0.252 (*p* < 0.001)	−0.226 (*p* < 0.001)
HADS-anxiety ^j^	0.400 (*p* < 0.001)	0.361 (*p* < 0.001)	0.359 (*p* < 0.001)
HADS-depression ^k^	0.360 (*p* < 0.001)	0.335 (*p* < 0.001)	0.322 (*p* < 0.001)
BMI ^l^	0.204 (*p* < 0.001)	0.127 (*p* < 0.001)	0.227 (*p* < 0.001)

Note: a: SF36 Physical functioning; b: SF36 Role physical; c: SF36 Bodily pain; d: SF36 General health; e: SF36 Vitality; f: SF36 Social Functioning; g: SF36 Role emotional; h: SF36 Mental Health; i: European Quality of Life-5 Dimensions’ Coefficient; j: Anxiety subscale of the HADS questionnaire; k: Depression subscale of the HADS questionnaire; l: Body mass index. (*p* < 0.001).

**Table 4 ijerph-19-10965-t004:** Exploratory factor analysis of the EEQ (*N =* 1282) (*p* < 0.001).

EEQ Items	Component 1	Component 2
1. Does the scale have a great power over you? Is it able to change your mood?	0.183	0.689
2. Do you tend to have a whim for certain foods?	0.591	0.231
3. Do you find it difficult to stop eating sweet foods, especially chocolate?	0.587	0.204
4. Do you have problems controlling the amounts of certain foods?	0.557	0.388
5. Do you eat when you are stressed, angry, or bored?	0.674	0.245
6. Do you eat more of your favourite foods, and eat more out of control, when you are alone?	0.650	0.196
7. Do you feel guilty when you eat forbidden foods, i.e., foods that you think you should not eat, such as sweets or snacks?	0.207	0.690
8. In the evening, when you come home tired from work, is it when you feel most out of control in your eating?	0.403	0.326
9. You are on a diet, and for some reason you eat more than you should, so ¿do you think it is not worth it and, therefore, you eat in an uncontrolled way those foods that you think will make you gain weight?	0.394	0.503
10. How often do you feel that food controls you instead of you controlling it?	0.457	0.656

**Table 5 ijerph-19-10965-t005:** Description of the different EEQ models and confirmatory factor analysis of three different models.

Model	x2-	DF	X^2^/DF	CFI	IFI	RMSEA
Model 1 (Garaulet et al., 2012) [24]	261.05	32	8.16	0.922	0.922	0.075
Model 2 (Bernabéu et al., 2020) [13]	119.42	33	3.62	0.970	0.971	0.045
Model 3 (proposed)	101.75	33	3.08	0.977	0.977	0.040

DF = degrees of freedom; CFI = comparative fit index; IFI = incremental fit index; RMSEA = root mean square error of approximation. (*p* < 0.001).

**Table 6 ijerph-19-10965-t006:** Reliability results of the EEQ items (*p* < 0.001).

	Corrected Item-Total Correlation	Cronbach’s Alpha If the Item Has Been Removed
1. Does the scale have a great power over you? Is it able to change your mood?	0.528	0.849
2. Do you tend to have a whim for certain foods?	0.575	0.845
3. Do you find it difficult to stop eating sweet foods, especially chocolate?	0.557	0.846
4. Do you have problems controlling the amounts of certain foods?	0.623	0.841
5. Do you eat when you are stressed, angry, or bored?	0.597	0.842
6. Do you eat more of your favourite foods, and eat more out of control, when you are alone?	0.577	0.844
7. Do you feel guilty when you eat forbidden foods, i.e., Foods that you think you should not eat, such as sweets or snacks?	0.525	0.849
8. In the evening, when you come home tired from work, is it when you feel most out of control in your eating?	0.444	0.855
9. You are on a diet, and for some reason you eat more than you should, so ¿do you think it is not worth it and, therefore, you eat in an uncontrolled way those foods that you think will make you gain weight?	0.580	0.845
10. How often do you feel that food controls you instead of you controlling it?	0.687	0.835

## Data Availability

The datasets used and/or analyzed during the current study are available from the corresponding author upon reasonable request.

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
