# Peer review of "Psychometric Properties of the Emotional Eater Questionnaire in University Students"

_ijerph, 2022, doi:10.3390/ijerph191710965_

Round 1

Reviewer 1 Report (New Reviewer)

The manuscript entitled ‘ Psychometric Properties of the Emotional Eater Questionnaire in University Students ‘presents interesting issue, however some corrections are needed. The manuscript is well prepared and worth to be published.  

-       There are no line numbers, I cannot write what lines have errors.

-       There are some typos (e.g. ‘ p<<0.0011)’

-       More information about Test-retest reliability should be presented. Taking into account the data was anonymous, how authors contact this this 152 students. What was a period between test and re-test.

-       Table 1 – it should be ‘sd= 4.81’ instead of ‘sd= 4,81’

-       Table 2 – there was a quite big floor effect - how can this be explained?

-       Table 5 – ‘propuesto’  - what does it mean?  

-       Figure 1 should be describe in results

-       The discussion section should be added. (The section entitled ‘4. Strengths and weaknesses of the survey’ is like a discussion – so authors should rename it, and add some more literature. Moreover, add the limitation section at the end of the discussion section or as a separate section.

-       In limitation section authors should add information that weight, and height were self-reported information (not really objective data). How it could influence the results?

Author Response

Reviewer 2 Report (New Reviewer)

Dear authors,

Thank you to give the opportunity to review your manuscript.

General

Bibliography in the manuscript should be in Vancouver style.

1. Introduction

A reference is needed for that affirmation.

“Estimates of the prevalence of EE vary across different studies. Approximately 20-45% of non-clinical adult samples are identified as emotional eaters.”

On the next paragraph, should be indicated the author of the study, for example: Garaulet et al, validated the questionnaire

It has been used to assess EE in a Spanish population with binge eating disorder. The validation of this questionnaire was carried out in 354 participants with obesity, all of them undergoing dietary and behavioural treatment for weight loss. After principal com-ponents analysis, the questionnaire was classified in three different dimensions that ex-plained 60% of the total variance: Disinhibition, Type-of-food and Guilt. Internal con-sistency showed that Cronbach's alpha was 0.773 for the "Dishinibition" subscale, 0.656 for the "Type of food" subscale and 0.612 for the "Guilt" subscale. The test-retest stability was r = 0.70. The

2.3. Data collection

Description of the methods should be included in the “2.2. Instrument” in the section

3.1. Description of the sample

SF36 were SF36VT, SF36MH, SF36GH, and SF36RE; HADS-D Should be explained to understand better the subcategories.

Tables: At the bottom of the table, it should be explained which statistical analysis has been used to establish significance. Apply in all the tables.

 Best regards, 

Author Response

This manuscript is a resubmission of an earlier submission. The following is a list of the peer review reports and author responses from that submission.

Round 1

Reviewer 1 Report

Thank you for the opportunity to review your work: Psychometric Properties of the Emotional Eater Questionnaire in University Students. The paper seems valuable because as the authors note: emotional eating patterns have been shown to play a relevant role in the development of overweight and obesity.

In the methodology please describe in detail the criteria for qualifying respondents and what methods were used to avoid falsification of data (fake/bot responders) - Respondent codes? Time of completion? Direct links?

Please have separate sections for the strengths and weaknesses of the survey and for the conclusion.

The initials (A.M.-O.G.) were stuck at the end of the proper text - please remove them. Also, in some places the font is larger than the editing requirements suggest.

I wish you all the best!

Author Response

Response to Reviewer 1 Comments

Point 1: In the methodology, please describe in detail the criteria for qualifying respondents and what methods were used to avoid falsification of data (fake/bot responders) - Respondent codes? Time of completion? Direct links?

Response 1: Thank you very much for your comment. We believe this revision have resulted in a significantly improved manuscript since the information you request is very important for the quality of the article and for the readers. For this reason, we have incorporated all the data according to your request in the methodology section (point 2.3).

Point 2: Please have separate sections for the strengths and weaknesses of the survey and for the conclusion.

Response 2: We agree with this and have incorporated your suggestion throughout the

Manuscript (point 4 &5).

Point 3: The initials (A.M.-O.G.) were stuck at the end of the proper text - please remove them. Also, in some places the font is larger than the editing requirements suggest.

Response 3: Thank you for pointing this out. We agree with this comment. Therefore, we have removed the initials (A.M.-O.G.) at the end of the proper text. Moreover, we have edited the font size according to the editing requirements.

Reviewer 2 Report

This study examines an important area of disordered eating that is attracting increasing interest from researchers. As such, the validation of instruments for measuring Emotional Eating is needed across different languages and contexts due to the wide cultural variation in eating behaviors around the world.  The authors need to be more careful to avoid secondary citations throughout. The manuscript requires reformatting/merging of the excessive amount of tables for better organization.

The specific comments are as follows:

Abstract:

1)      Please state where Huelva is (i.e. what country), as very few readers will have heard of this city.

2)      Please state when the study was conducted (give months and year).

3)      Please explicitly state whether the EEQ questionnaire was in Spanish (this needs to be explicitly stated).

4)      For reporting of Cronbach’s alpha, limit to decimal places 0.687-à 0. 69. One decimal point for age, and BMI.

Introduction:

1)      The first reference cites commonly known concept (cumulative life-course effects on long-term health and mortality), the authors can cite more authoritative studies on this:

Batty GD, Hamer M, Gale CR. Life-course Psychological Distress and Total Mortality by Middle Age: The 1970 Birth Cohort Study. Epidemiology. 2021;32(5):740-743. doi:10.1097/EDE.0000000000001374

2)      Paragraph 5, there are many studies about positive emotional eating that can be cited in addition to Cardi et al., 2015).  The sentence states” There are now a number of studies showing that positive mood can also lead to increased food intake (Cardi et al., 2015)”.  But only 1 study is cited—please insert the other missing citations eg; Bongers, P., de Graaff, A., and Jansen, A. (2016). ‘Emotional’ does not even start to cover it: generalization of overeating in emotional eaters. Appetite 96, 611–616. doi: 10.1016/j.appet.2015.11.004  and other studies.

3)      Paragraph 7 of the introduction cites prevalence of EE in the general population. But for comparability, the introduction should also cite studies of university students from various parts of the world, including Western populations and non-Western populations to give the reader a general sense of extent of this problem, globally. E.g.:

Sze, K.Y.P., Lee, E.K.P., Chan, R.H.W. et al. Prevalence of negative emotional eating and its associated psychosocial factors among urban Chinese undergraduates in Hong Kong: a cross-sectional study. BMC Public Health 21, 583 (2021). https://doi.org/10.1186/s12889-021-10531-3.

Sultson H, Kukk K, Akkermann K. Positive and negative emotional eating have different associations with overeating and binge eating: Construction and validation of the Positive-Negative Emotional Eating Scale. Appetite. 2017;116:423-430. doi:10.1016/j.appet.2017.05.035

Tariq A. Alalwan, Sawsan J. Hilal, Alaa M. Mahdi, Maryam A. Ahmed & Qaher A. Mandeel (2019) Emotional eating behavior among University of Bahrain students: a cross-sectional study, Arab Journal of Basic and Applied Sciences, 26:1, 424-432, DOI: 10.1080/25765299.2019.1655836

4)      Paragraph 8, first sentence “Negative emotions are an integral part of life (Evers et al., 2010)” should not be cited. It is strange to cite something that is so well-acknowledged. May remove this sentence.

5)      The Introduction is extremely long [suggest to merge and streamline paragraphs 4, 5 & 6]

6)      Paragraphs 10-12: unclear what knowledge gap this study is addressing—the Garaulet et al., 2012 conducted a validation: state the results of that & Gonzalez, 2018 should have reported some psychometric aspects.  Please state more clearly whether only exploratory factor analysis had been done before.

Methods:

7)      The authors should state the country of this study since the readers will be unlikely know where University of Huelva is.  If in Spain, please make brief mention as it’s general location (e.g. eastern coast) and whether it is a large or medium-sized city or rural.

8)      Please give citation for SPSS software (all software should be cited).

9)      “with Kappa index of <0.40… indicating adequate reliability of the instrument--- please cite the correct citation. The authors seem to be doing secondary citations (citing a statement from a paper that states this) rather than citing the appropriate statistical/psychometric reference that tests Kappa statistics. Please cite the appropriate citation for Kappa. Citing Gauralet et al. is not appropriate as the study is about EEQ not Kappa statistics. A more appropriate citation would be (please make sure to avoid secondary citaitons) throughout.

McHugh ML. Interrater reliability: the kappa statistic. Biochem Med (Zagreb). 2012;22(3):276-282.

10)  The EFA and CFA of the newly proposed model should be conducted with split-half samples of the study population (it does not make sense to conduct EFA on the 1282 students and to rerun on the same sample a CFA). The sample must be randomly split and EFA conducted on 641 and CFA conducted on the remaining sample to confirm the factor structure, particularly for the 3rd model.  (Or the authors can run the EFA on the 1282 and recruit another sample to run the CFA).

Results: 

11)  The reference for Bentler appears to be misformatted or incomplete in the in-text citation and reference list. The ampersand seems to indicated 2 authors.

12)  Please merge tables 2 & 3 (they are mainly descriptive of the study sample and should be streamlined and merged together with data stratified by gender into 4 columns: (Column 1: Males, Column 2: Females,  Column 3: Chi-square or t-test p-values and Column 4: total sample).

13)  Please merge the contents of Table 1 into the contents of Table 9—they should be together for clarity.

14)  Table 6 is unclear: The Results section state that the overall Cronbach’s alpha is 0.859 and 0.805 for dimension 1 and 0.787 for dimension 2—but does not specify which model they are referring to (their own proposed or one or Model 1 or 2). And the

15)  Table 7 is unnecessary and the authors can simply report as they did in the results that the variance of the 2 factors with eigenvalues >1 was 56% (44.7% for the first factor and 11.3% for the second).

16)   For the EFA results, please give some sort of name for Factor 1 (e.g. Impulsive Eating) and Factor 2 (Mood/Affecct)  to tell the reader what these factors are likely measuring.

            General Comment for Discussion:

17)  The Discussion has extensive discussion about the variations in factor structure of the EEQ across different populations (paragraph 3) which can be streamlined (remove mention of all item numbers as it is tedious to read and have readers instead refer to Table 1). As all the 3 models did not drop any of the items, the entire EEQ remains intact and the only difference between models 2 & 3 is item #9’s inclusion under different factors. The main discussion of the factor structure should therefore focus on the difference between Gaurelet vs the other 2 models (3 factor versus 2 factor)—these factors should be described conceptually—what were Gaurelet’s 3 factors seeming to refer to?

18)  The discussion should put more emphasis on the validation study—as to the differences found between this analysis and validation from other studies—what were the differences or similarities with the Bernabeu et al? Did the findings of this study largely confirm any validation studies conducted previously?

Author Response

Response to Reviewer 2 Comments

Point 1: The first reference cites commonly known concept (cumulative life-course effects on long-term health and mortality), the authors can cite more authoritative studies on this: Batty GD, Hamer M, Gale CR. Life-course Psychological Distress and Total Mortality by Middle Age: The 1970 Birth Cohort Study. Epidemiology. 2021;32(5):740-743. doi:10.1097/EDE.0000000000001374

Response 1: Our sincere appreciation. We agree with this and have cited a more authoritative study in the manuscript (Introduction paragraph 1).

Point 2: Paragraph 5, there are many studies about positive emotional eating that can be cited in addition to Cardi et al., 2015).  The sentence states” There are now a number of studies showing that positive mood can also lead to increased food intake (Cardi et al., 2015)”.  But only 1 study is cited—please insert the other missing citations eg; Bongers, P., de Graaff, A., and Jansen, A. (2016). ‘Emotional’ does not even start to cover it: generalization of overeating in emotional eaters. Appetite 96, 611–616. doi: 10.1016/j.appet.2015.11.004  and other studies.

Response 2: Please accept our deepest thanks. That is right so we have cited some more studies about positive emotional eating as (Bongers et al., 2016; Sultson et al., 2017) in the manuscript (Introduction paragraph 4).

Point 3: Paragraph 7 of the introduction cites prevalence of EE in the general population. But for comparability, the introduction should also cite studies of university students from various parts of the world, including Western populations and non-Western populations to give the reader a general sense of extent of this problem, globally.

Response 3: Thank you for your guidance. We concur with you and we have cited studies related to prevalence of EE in the general population as Alalwan et al., 2019; Sultson et al., 2017; Sze et al., 2021 (Introduction paragraph 5).

Point 4: Paragraph 8, first sentence “Negative emotions are an integral part of life (Evers et al., 2010)” should not be cited. It is strange to cite something that is so well-acknowledged. May remove this sentence.

Response 4: Thank you for your support. We could not agree with you more, so we have removed this sentence.

Point 5: The Introduction is extremely long [suggest merging and streamline paragraphs 4, 5 & 6]

Response 5: Our sincere thanks. There is no doubt, we have merged and streamlined paragraphs 4, 5 & 6 as suggested.

Point 6: Paragraphs 10-12: unclear what knowledge gap this study is addressing—the Garaulet et al., 2012 conducted a validation: state the results of that & Gonzalez, 2018 should have reported some psychometric aspects.  Please state more clearly whether only exploratory factor analysis had been done before.

Response 6: Our sincere gratitude. That is absolutely true. We have stated the available results of Garaulet and Gonzalez and state more clearly that only EFA had been done before (Introduction paragraph 10 & 11).

Point 7: The authors should state the country of this study since the readers will be unlikely know where University of Huelva is.  If in Spain, please make brief mention as it’s general location (e.g. eastern coast) and whether it is a large or medium-sized city or rural.

Response 7: Thank you for your assistance. We agree with this and have said where Huelva is and that is a small town in the west coast of Spain (Material & Methods, Participants, paragraph 1).

Point 8: Please give citation for SPSS software (all software should be cited).

Response 8: Our thanks and appreciation. That is totally true, we have cited SPSS software version 25 (Material & Methods, Data collection, paragraph 1).

Point 9: “with Kappa index of <0.40… indicating adequate reliability of the instrument--- please cite the correct citation. The authors seem to be doing secondary citations (citing a statement from a paper that states this) rather than citing the appropriate statistical/psychometric reference that tests Kappa statistics. Please cite the appropriate citation for Kappa. Citing Gauralet et al. is not appropriate as the study is about EEQ not Kappa statistics. A more appropriate citation would be (please make sure to avoid secondary citaitons) throughout. McHugh ML. Interrater reliability: the kappa statistic. Biochem Med (Zagreb). 2012;22(3):276-282.

Response 9: Thank you for your time. You stand unopposed, we have cited a study about kappa statistic (Introduction paragraph 10).

Point 10: The EFA and CFA of the newly proposed model should be conducted with split-half samples of the study population (it does not make sense to conduct EFA on the 1282 students and to rerun on the same sample a CFA). The sample must be randomly split and EFA conducted on 641 and CFA conducted on the remaining sample to confirm the factor structure, particularly for the 3rd model.  (Or the authors can run the EFA on the 1282 and recruit another sample to run the CFA).

Response 10: Please accept our deepest thanks. We have randomly splitted-half samples and we have conducted the EFA on 641 and CFA on the remaining sample to confirm the factor structure for all the models (point 3.4.3 & 3.4.4.). This is explained in point 2.4. paragraph 3.

Point 11: The reference for Bentler appears to be misformatted or incomplete in the in-text citation and reference list. The ampersand seems to indicated 2 authors.

Response 11: Our sincere gratitude, we have reviewed and corrected the reference for Bentler (point 3.4.4 paragraph 1).

Point 12: Please merge tables 2 & 3 (they are mainly descriptive of the study sample and should be streamlined and merged together with data stratified by gender into 4 columns: (Column 1: Males, Column 2: Females, Column 3: Chi-square or t-test p-values and Column 4: total sample).

Response 12: Thank you for your guidance. You are totally right; we have merged tables 2 & 3 as suggested (now table 2).

Point 13: Please merge the contents of Table 1 into the contents of Table 9—they should be together for clarity.

Response 13: Our sincere appreciation. That´s totally true, we have merged tables 1 & 9 (now table 6).

Point 14: Table 6 is unclear: The Results section state that the overall Cronbach’s alpha is 0.859 and 0.805 for dimension 1 and 0.787 for dimension 2—but does not specify which model they are referring to (their own proposed or one or Model 1 or 2).

Response 14: Thank you for your assistance. There is no better way to say that we have specified that the overall Cronbach’s alpha is 0.859 and 0.805 for dimension 1 and 0.787 for dimension 2 is referring to our proposed Model (point 3.4.1, paragraph 1).

Point 15: Table 7 is unnecessary, and the authors can simply report as they did in the results that the variance of the 2 factors with eigenvalues >1 was 56% (44.7% for the first factor and 11.3% for the second).

Response 15: Our thanks and appreciation. We agree and have removed the table 7.

Point 16: For the EFA results, please give some sort of name for Factor 1 (e.g. Impulsive Eating) and Factor 2 (Mood/Affecct) to tell the reader what these factors are likely measuring.

Response 16: Our sincere thanks. That´s undeniably true so we have given these names to the factors in the Figure 1 legend.

Point 17: The Discussion has extensive discussion about the variations in factor structure of the EEQ across different populations (paragraph 3) which can be streamlined (remove mention of all item numbers as it is tedious to read and have readers instead refer to Table 1). As all the 3 models did not drop any of the items, the entire EEQ remains intact and the only difference between models 2 & 3 is item #9’s inclusion under different factors. The main discussion of the factor structure should therefore focus on the difference between Gaurelet vs the other 2 models (3 factor versus 2 factor)—these factors should be described conceptually—what were Gaurelet’s 3 factors seeming to refer to?

Response 17: Thank you for your support. That´s a really good point. We have summarised paragraph 3 and focus it on the difference between Gaurelet vs the other 2 models (Discussion, paragraph 4).

Point 18: The discussion should put more emphasis on the validation study—as to the differences found between this analysis and validation from other studies—what were the differences or similarities with the Bernabeu et al? Did the findings of this study largely confirm any validation studies conducted previously?

Response 18: Thank you for your time, we agree with you and have specified the differences with Bernabeu et al and stated that this study confirm any validation studies previously (Discussion, paragraph 6).

Round 2

Reviewer 2 Report

The manuscript is considerably improved.